# Powerpropagation:
# A sparsity inducing weight reparameterisation

**Jonathan Schwarz**
DeepMind &
Gatsby Unit, UCL
schwarzjn@google.com

**Siddhant M. Jayakumar**
DeepMind &
University College London

**Razvan Pascanu**
DeepMind

**Peter E. Latham**
Gatsby Unit, UCL

**Yee Whye Teh**
DeepMind

## Abstract

The training of sparse neural networks is becoming an increasingly important tool for reducing the computational footprint of models at training and evaluation, as well enabling the effective scaling up of models. Whereas much work over the years has been dedicated to specialised pruning techniques, little attention has been paid to the inherent effect of gradient based training on model sparsity. In this work, we introduce Powerpropagation, a new weight-parameterisation for neural networks that leads to *inherently sparse* models. Exploiting the behaviour of gradient descent, our method gives rise to weight updates exhibiting a "rich get richer" dynamic, leaving low-magnitude parameters largely unaffected by learning. Models trained in this manner exhibit similar performance, but have a distribution with markedly higher density at zero, allowing more parameters to be pruned safely. Powerpropagation is general, intuitive, cheap and straight-forward to implement and can readily be combined with various other techniques. To highlight its versatility, we explore it in two very different settings: Firstly, following a recent line of work, we investigate its effect on sparse training for resource-constrained settings. Here, we combine Powerpropagation with a traditional weight-pruning technique as well as recent state-of-the-art sparse-to-sparse algorithms, showing superior performance on the ImageNet benchmark. Secondly, we advocate the use of sparsity in overcoming catastrophic forgetting, where compressed representations allow accommodating a large number of tasks at fixed model capacity. In all cases our reparameterisation considerably increases the efficacy of the off-the-shelf methods.

## 1 Introduction

Deep learning models are emerging as the dominant choice across several domains, from language [e.g. 1, 2] to vision [e.g. 3, 4] to RL [e.g. 5, 6]. One particular characteristic of these architectures is that they perform optimally in the overparameterised regime. In fact, their size seems to be mostly limited by hardware constraints. While this is potentially counter-intuitive, given a classical view on overfitting, the current understanding is that model size tends to have a dual role: It leads to better behaved loss surfaces, making optimisation easy, but also acts as a regulariser. This gives rise to the *double descent* phenomena, where test error initially behaves as expected, growing with model size due to overfitting, but then decreases again as the model keeps growing, and asymptotes as the model size goes to infinity to a better performance than obtained in the classical regime [7, 8].

35th Conference on Neural Information Processing Systems (NeurIPS 2021).

As a consequence, many of the state of the art models tend to be prohibitively large, making them inaccessible to large portions of the research community despite scale still being a driving force in obtaining better performance. To address the high computational cost of inference, a growing body of work has been exploring ways to compress these models. As highlighted by several works [e.g. 9, 10, 11], size is only used as a crutch during the optimisation process, while the final solution requires a fraction of the capacity of the model. A typical approach therefore is to sparsify or prune the neural network after training by eliminating parameters that do not play a vital role in the functional behaviour of the model. Furthermore, there is a growing interest in sparse training [e.g. 12, 13, 14], where the model is regularly pruned or sparsified during training in order to reduce the computational burden.

Compressed or sparse representations are not merely useful to reduce computation. Continual learning, for example, focuses on learning algorithms that operate on non-iid data [e.g. 15, 16, 17]. In this setup, training proceeds sequentially on a set of tasks. The system is expected to accelerate learning on subsequent tasks as well as using newly acquired knowledge to potentially improve on previous problems, all of this while maintaining low memory and computational footprint. This is difficult due to the well studied problem of catastrophic forgetting, where performance on previous tasks deteriorates rapidly when new knowledge is incorporated. Many approaches to this problem require identifying the underlying set of parameters needed to encode the solution to a task in order to freeze or protect them via some form of regularisation. In such a scenario, given constraints on model size, computation and memory, it is advantageous that each learned task occupies as little capacity as possible.

In both scenarios, typically, the share of capacity needed to encode the solution is determined by the learning process itself, with no explicit force to impose frugality. This is in contrast with earlier works on $L_0$ regularisation that explicitly restrict the learning process to result in sparse and compressible representations [18]. The focus of our work, similar to the $L_0$ literature, is on how to encourage the learning process to be frugal in terms of parameter usage. However instead of achieving this by adding an explicit penalty, we enhance the "rich get richer" nature of gradient descent. In particular we propose a new parameterisation that ensures steps taken by gradient descent are proportional to the magnitude of the parameters. In other words, parameters with larger magnitudes are allowed to adapt faster in order to represent the required features to solve the task, while smaller magnitude parameters are restricted, making it more likely that they will be irrelevant in representing the learned solution.

## 2    Powerpropagation

The desired proportionality of updates to weight magnitudes can be achieved in a surprisingly simple fashion: In the forward pass of a neural networks, raise the parameters of your model (element-wise) to the $\alpha$-th power (where $\alpha > 1$) while preserving the sign. It is easy to see that due to the chain rule of calculus the magnitude of the parameters (raised to $\alpha - 1$) will appear in the gradient computation, scaling the usual update. Therefore, small magnitude parameters receive smaller gradient updates, while larger magnitude parameters receive larger updates, leading to the aforementioned "rich get richer" phenomenon. This simple intuition leads to the name of our method.

More formally, we enforce sparsity through an implicit regulariser that results from reparameterising the model similar to [19, 20]. This line of research builds on previous work on matrix factorisation [e.g. 21, 22]. In [19] a parameterisation of the form $w = v \odot v - u \odot u$ is used to induce sparsity, where $\odot$ stands for element-wise multiplication and we need both $v$ and $u$ to be able to represent negative values, since the parameters are effectively squared. In our work we rely on a simpler formulation where $w = v|v|^{\alpha-1}$, for any arbitrary power $\alpha \geq 1$ (as compared to fixing $\alpha$ to 2), which, since we preserved the sign of $v$, can represent both negative and positive values. For $\alpha = 1$ this recovers the standard setting.

If we denote by $\Theta = \mathcal{R}^M$ the original parameter space or manifold embedded in $\mathcal{R}^M$, our reparameterisation can be understood through an invertible map $\Psi$, where its inverse $\Psi^{-1}$ projects $\theta \in \Theta$ into $\phi \in \Phi$, where $\Phi$ is a new parameter space also embedded in $\mathcal{R}^M$, i.e. $\Phi = \mathcal{R}^M$. The map is defined by applying the function $\Psi : \mathcal{R} \rightarrow \mathcal{R}, \Psi(x) = x|x|^{\alpha-1}$ element-wise, where by abuse of notation we refer to both the vector and element level function by $\Psi$. This new parameter space or manifold $\Phi$ has a curvature (or metric) that depends on the Jacobian of $\Psi$. Similar constructions have

been previously used in optimisation, as for example in the case of the widely known Mirror Descent algorithm [23], where the invertible map $\Psi$ is the link function. For deep learning, Natural Neural Networks [24] rely on a reparameterisation of the model such that in the new parameter space, at least initially, the curvature is close to the identity matrix, making a gradient descent step similar to a second order step. Warp Gradient Descent [25] relies on a meta-learning framework to learn a nonlinear projection of the parameters with a similar focus of improving efficiency of learning. In contrast, our focus is to construct a parameterisation that leads to an implicit regularisation towards sparse representation, following [19], rather then improving convergence.

Given the form of our mapping $\Psi$, in the new parameterisation the original weight $\theta_i$ will be replaced by $\phi_i$, where $\Psi(\phi_i) = \phi_i|\phi_i|^{\alpha-1} = \theta_i$ and $i$ indexes over the dimensionality of the parameters. Note that we apply this transformation only to the weights of a neural network, leaving other parameters untouched. Given the reparameterised loss $\mathcal{L}(\cdot, \Psi(\phi))$, the gradient wrt. to $\phi$ becomes

$$\frac{\partial \mathcal{L}(\cdot, \Psi(\phi))}{\partial \phi} = \frac{\partial \mathcal{L}}{\partial \Psi(\phi)} \frac{\partial \Psi(\phi)}{\partial \phi} = \frac{\partial \mathcal{L}}{\partial \Psi(\phi)} \text{diag}(\alpha|\phi|^{\circ\alpha-1}). \tag{1}$$

Note that $\text{diag}$ indicates a diagonal matrix, and $|\phi|^{\circ\alpha-1}$ indicates raising element-wise the entries of vector $|\phi|$ to the power $\alpha - 1$. $\frac{\partial \mathcal{L}}{\partial \Psi(\phi)}$ is the derivative wrt. to the original weight $\theta = \phi|\phi|^{\circ\alpha-1}$ which is the gradient in the original parameterisation of the model. This is additionally multiplied (element-wise) by the factor $\alpha|\phi_i|^{\alpha-1}$, which will scale the step taken proportionally to the magnitude of each entry. Finally, for clarity, this update is different from simply scaling the gradients in the original parameterisation by the magnitude of the parameter (raised at $\alpha - 1$), since the update is applied to $\phi$ not $\theta$, and is scaled by $\phi$. The update rule (1) has the following properties:

(i) 0 is a critical point for the dynamics of any weight $\phi_i$, if $\alpha > 1$. This is easy to see as $\frac{\partial \mathcal{L}}{\partial \phi_i} = 0$ whenever $\phi_i = 0$ due to the $\alpha|\phi_i|^{\alpha-1}$ factor.

(ii) In addition, 0 is surrounded by a plateau and hence weights are less likely to change sign (gradients become vanishingly small in the neighbourhood of 0 due to the scaling). This should negatively affect initialisations that allow for both negative and positive values, but it might have bigger implications for biases.

(iii) This update is naturally obtained by the Backpropagation algorithm. This comes from the fact that Backpropagation implies applying the chain-rule from the output towards the variable of interest, and our reparameterisation simply adds another composition (step) in the chain before the variable of interest.

At this point the perceptive reader might be concerned about the effect of equation (1) on established practises in the training of deep neural networks. Firstly, an important aspect of reliable training is the initialisation ([e.g. 26, 27, 28]) or even normalisation layers such as batch-norm [29] or layer-norm [30]. We argue that our reparameterisation preserves all properties of typical initialisation schemes as it does not change the function. Specifically, let $\theta_i \sim p(\theta)$ where $p(\theta)$ is any distribution of choice. Then our reparameterisation involves initialising $\phi_i \leftarrow \text{sign}(\theta_i) \cdot \sqrt[\alpha]{|\theta_i|}$, ensuring the neural network and all intermediary layers are functionally the same. This implies that hidden activations will have similar variance and mean as in the original parameterisation, which is what initialisation and normalisation focus on.

Secondly, one valid question is the impact of modern optimisation algorithms ([e.g. 31, 32, 33]) on our reparameterisation. These approaches correct the gradient step by some approximation of the curvature, typically given by the square root of a running average of squared gradients. This quantity will be proportional (at least approximately) to $\text{diag}(\alpha|\phi|^{\circ\alpha-1})^1$. This reflects the fact that our projection relies on making the space more curved and implicitly optimisation harder, which is what these optimisation algorithms aim to fix. Therefore, a naive use with Powerprop. would result in a reduction of the "rich get richer" effect. On the other hand, avoiding such optimisers completely can considerably harm convergence and performance. The reason for this is that they do not only

---

[1]To see this assume the weights do not change from iteration to iteration. Then each gradient is scaled by the same value $\text{diag}(\alpha|\phi|^{\circ2(\alpha-1)})$ which factors out in the summation of gradients squared, hence the correction from the optimiser will undo this scaling. In practice $\phi$ changes over time, though slowly, hence approximately this will still hold.

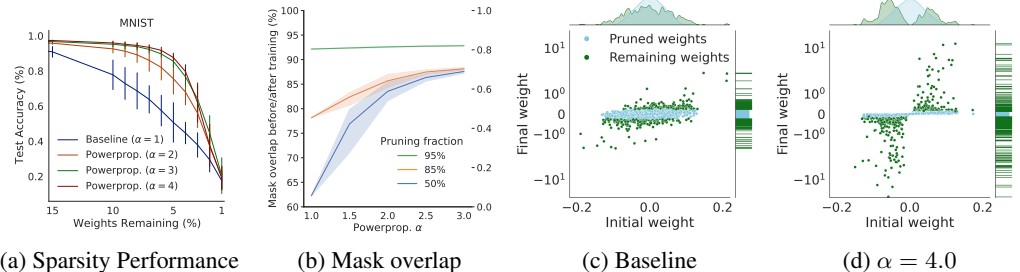

(a) Sparsity Performance    (b) Mask overlap    (c) Baseline    (d) $\alpha = 4.0$

Figure 1: Powerpropagation applied to Image classification. (a) Test accuracy at increasing levels of sparsity for MNIST (b) Overlap between masks computed before and after training (c) & (d) Analysis of weight distributions for a Baseline model and at a high $\alpha$. We use 10K weights chosen at random from the network. For a) & b) we show mean and standard deviation over 5 runs. *We provide code to reproduce the MNIST results (a) in the accompanying notebook.*

correct for the curvature induced by our reparameterisation, but also the intrinsic curvature of the problem being solved. Removing the second effect can make optimisation very difficult. We provide empirical evidence of this effect in the Appendix.

To mitigate this issue and make Powerprop. straightforward to use in any setting, we take inspiration from the target propagation literature [34, 35, 36, 37] which proposes an alternative way of thinking about the Backpropagation algorithm.

In our case, we pretend that the exponentiated parameters are the de-facto parameters of the model and compute an update wrt. to them using our optimiser of choice. The updated exponentiated parameters are then seen as a *virtual target*, and we take a gradient descent step on $\phi$ towards these virtual targets. This will result in a descent step, which, while it relies on modern optimisers to correct for the curvature of the problem, does not correct for the curvature introduced by our parameterisation. Namely if we denote $optim : \mathcal{R}^M \to \mathcal{R}^M$ as the function that implements the correction to the gradient done by some modern optimiser, our update becomes $\Delta\phi = optim\left(\frac{\partial\mathcal{L}}{\partial\Psi(\phi)}\right)\text{diag}(\alpha|\phi|^{\circ\alpha-1})$.

The proof that our update is indeed correct follows the typical steps taken in the target propagation literature. From a first order Taylor expansion of $\mathcal{L}(\phi - \eta\Delta\phi)$, we have that in order for $\Delta\phi$ to reduce the loss, the following needs to hold: $\langle\Delta\phi, \frac{\partial\mathcal{L}}{\partial\phi}\rangle > 0$. But we know that $\left\langle optim\left(\frac{\partial\mathcal{L}}{\partial\Psi(\phi)}\right), \frac{\partial\mathcal{L}}{\partial\Psi(\phi)}\right\rangle > 0$ as this was a valid step on $\Psi(\phi)$. Because $\text{diag}(\alpha|\phi|^{\circ\alpha-1})$ is positive definite (diagonal matrix with all entries positive), we can multiply it on both sides, proving that $\langle\Delta\phi, \frac{\partial\mathcal{L}}{\partial\phi}\rangle > 0$. We provide more details in the Appendix. We will rely on this formulation in our empirical evaluation.

## 3    Effect on weight distribution and sparsification

At this point an empirical demonstration might illuminate the effect of equation (1) on model parameters and our ability to sparsify them. Throughout this work, we will present results from neural networks after the removal of low-magnitude weights. We prune such parameters by magnitude (i.e. $\min|\theta_i|$), following current best practice [e.g. 38, 39, 13]. This is based on a Taylor expansion argument [14] of a sparsely-parameterised function $f(x, \theta_s)$ which we would like to approximate its dense counterpart $f(x, \theta)$: $f(x, \theta_s) \approx f(\theta, x) + g^T(\theta_s - \theta) + \frac{1}{2}(\theta_s - \theta)^T H(\theta_s - \theta) + ...$ where $g$ is the gradient vector and $H$ the Hessian. As higher order derivatives are impractical to compute for modern networks, minimising the norm of $(\theta_s - \theta)$ is a practical choice instead.

Following the experimental setup in [10] we study the effect of Powerpropagation at different powers ($\alpha$) relative to standard Backpropagation with otherwise identical settings. Figure 1a shows the effect of increasing sparsity on the layerwise magnitude-pruning setting for LeNet [40] on MNIST [41]. In both cases we notice a significant improvement over an otherwise equivalent baseline. While the choice of $\alpha$ does influence results, all choices lead to an improvement[2] in the MNIST setting. Where does this improvement come from? Figures 1c & 1d compare weights before and after training

---

[2]For numerical stability reasons we typically suggest a choice of $\alpha \in (1, 3]$ depending on the problem.

on MNIST at identical initialisation. We prune the network to 90% and compare the respective weight distributions. Three distinct differences are particularly noticeable: (i) Most importantly, weights initialised close to zero are significantly less likely to survive the pruning process when Powerpropagation is applied (see green Kernel density estimate). (ii) Powerpropagation leads to a heavy-tailed distribution of trained weights, as (1) amplifies the magnitude of such values. These observations are what we refer to as the "rich-get-richer" dynamic of Powerpropagation. (iii) Weights are less likely to change sign (see Figure 1d), as mentioned in Section 2.

One possible concern of (i) and (ii) are that Powerpropagation leads to training procedures where small weights cannot escape pruning, i.e. masks computed at initialisation and convergence are identical. This is undesirable as it is well established that pruning at initialisation is inferior [e.g. 11, 42, 43, 39]. To investigate this we plot the overlap between these masks at different pruning thresholds in Figure 1b. While overlap does increase with $\alpha$, at no point do we observe an inability of small weights to escape pruning, alleviating this concern. Code for this motivational example on MNIST is provided. [3]

## 4   Powerpropagation for Continual Learning

While algorithms for neural network sparsity are well established as a means to reduce training and inference time, we now formalise our argument that such advances can also lead to significant improvements in the continual learning setting: the sequential learning of tasks without forgetting. As this is an inherently resource constraint problem, we argue that many existing algorithms in the literature can be understood as implementing explicit or implicit forms of sparsity. One class of examples are based on weight-space regularisation [e.g. 17, 44] which can be understood as compressing the knowledge of a specific task to a small set of parameters that are forced to remain close to their optimal values. Experience Replay and Coreset approaches [e.g. 45, 46, 47] on the other hand compress data from previous tasks to optimal sparse subsets. The class of methods on which we will base the use of Powerpropagation for Continual Learning on implement gradient sparsity [e.g. 48, 49], i.e. they overcome catastrophic forgetting by explicitly masking gradients to parameters found to constitute the solution to previous tasks.

In particular, let us examine PackNet [48] as a representative of such algorithms. Its underlying principle is simple yet effective: Identify the subnetwork for each task through (iterative) pruning to a fixed budget, then fix the solution for each task by explicitly storing a mask at task switches and protect each such subnetwork by masking gradients from future tasks (using a backward Mask $\mathcal{M}^b$). Given a pre-defined number of tasks $T$, PackNet reserves $1/T$ of the weights. Self-evidently, this procedure benefits from networks that maintain high performance at increased sparsity, which becomes particularly important for large $T$. Thus, the application of improved sparsity algorithms such as Powerpropagation are a natural choice. Moreover, PackNet has the attractive property of merely requiring the storage of a binary mask per task, which comes at a cost of 1 bit per parameter, in stark contrast to methods involving the expensive storage (or generative modelling) of data for each past task.

Nevertheless, the approach suffers from its assumption of a known number of maximum tasks $T$ and its possibly inefficient resource allocation: By reserving a fixed fraction of weights for each task, no distinction is made in terms of difficulty or relatedness to previous data. We overcome both issues through simple yet effective modifications resulting in a method we term *EfficientPacknet* (EPN), shown in Algorithm 1. The key steps common to both methods are (i) Task switch (Line 3), (ii) Training through gradient masking (Line 4), (iii) Pruning (Line 8) , (iv) Updates to the backward mask needed to implement gradient sparsity.

Improvements of EPN upon PackNet are: **(a) Lines 7-11**: We propose a simple search over a range of sparsity rates $S = [s_1, \ldots, s_n]$, terminating the search once the sparse model's performance falls short of a minimum accepted target performance $\gamma P$ (computed on a held-out validation set) or once a minimal acceptable sparsity is reached. While the choice of $\gamma$ may appear difficult at first, we argue that an maximal acceptable loss in performance is often a natural requirement of a practical engineering application. In addition, in cases where the sparsity rates are difficult to set, a computationally efficient binary search (for $\gamma P$) up to a fixed number of steps can be performed instead. Finally, in smaller models the cost of this step may be reduced further by instead using

---

[3]`https://github.com/deepmind/deepmind-research/tree/master/powerpropagation`

**Algorithm 1:** EfficientPackNet (EPN) + Powerpropagation.

---

**Require:** $T$ tasks $[(X_1, y_1), \ldots, (X_T, y_T)]$; Loss & Eval. functions $\mathcal{L}, \mathcal{E}$; Initial weight distribution $p(u)$ (e.g. Uniform); $\alpha$ (for
Powerprop.); Target performance $\gamma \in [0, 1]$; Sparsity rates $S = [s_1, \ldots, s_n]$ where $s_{i+1} > s_i$ and $s_i \in [0, 1)$.

**Output :** Trained model $\phi$; Task-specific Masks $\{\mathcal{M}^t\}$

**1** $\mathcal{M}_i^b \leftarrow 1 \, \forall i$ // `Backward mask`

**2** $\phi_i \leftarrow \text{sign}(\theta) \cdot \sqrt[\alpha]{|\theta_i|}; \theta_i \sim p(\theta)$ // `Initialise parameters`

**3 for** $t \in [1, \ldots, T]$ **do**

**4**     $\phi \leftarrow \arg\min_\phi \mathcal{L}(X_t, y_t, \phi, \mathcal{M}_b)$ // `Train on task t with explicit gradient masking through` $\mathcal{M}_b$

**5**     $P \leftarrow \mathcal{E}(X_t, y_t, \phi)$ // `Validation performance of dense model on task t`

**6**     $l \leftarrow n$

**7**     **do**

       // `TopK(x, K) returns the indices of the K largest elements in a vector x`

**8**        $\mathcal{M}_i^t = \left\{ \begin{array}{ll} 1 & \text{if } i \in \text{TopK}(\phi, \lfloor s_l \cdot \dim(\phi) \rfloor) \\ 0 & \text{otherwise} \end{array} \right\}$ // `Find new Forward mask at sparsity` $s_l$

**9**        $P_s \leftarrow \mathcal{E}(X_T, y_T, \phi \odot \mathcal{M}^t)$ // `Validation performance of sparse model`

**10**        $l \leftarrow l - 1$

**11**     **while** $P_s > \gamma P \wedge l \geq 1$;

**12**     $\mathcal{M}^b \leftarrow \neg \bigvee_{i=1}^{t} \mathcal{M}^t$ // `Update backward mask to protect all tasks 1, ..., t`

**13**     Re-initialise pruned weights

       // `Optionally retrain with masked weights` $\phi \odot \mathcal{M}^t$ `on` $X_t, y_t$ `before task switch`

**14 end**

---

an error estimate based on a Taylor expansion [50]. This helps overcome the inefficient resource allocation of PackNet. **(b) Line 8**: More subtly, we choose the mask for a certain task among all network parameters, including ones used by previous tasks, thus encouraging reuse to existing parameters. PackNet instead forces the use of a fixed number of new parameters, thus possibly requiring more parameters than needed. Thus, the fraction of newly masked weights per task is adaptive to task complexity. **(c) Line 13**: We re-initialise the weights as opposed to leaving them at zero, due to the critical point property (Section 2). While we could leave the weight at their previous value, we found this to lead to slightly worse performance.

Together, these changes make the algorithm more suitable for long sequences of tasks, as we will show in our experiments. In addition, they overcome the assumption of an a priori known number of tasks $T$. Another beneficial property of the algorithm is that as the backward pass becomes increasingly sparse, the method becomes more computationally efficient with larger $T$. A possible concern for the method presented thus far is the requirement of known task ids at inference time, which are needed to select the correct mask for inputs. We present an algorithm to address this concern in the supplementary material.

## 5 Related work

Sparsity in Deep Learning has been an active research since at least the first wave of interest in neural networks following Backpropagation [51]. Early work was based on the Hessian [e.g. 52, 50, 53], an attractive but impractical approach in modern architectures. The popularity of magnitude-based pruning dates back to at least [54], whereas iterative pruning was first noted to be effective in [55]. Modern approaches are further categorised as Dense→Sparse or Sparse→Sparse methods, where the first category refers to the instantiation of a dense network that is sparsified throughout the training. Sparse→Sparse algorithms on the other hand maintain constant sparsity throughout, giving them a clear computational advantage.

Among Dense→Sparse methods, $L_0$ regularisation [18] uses non-negative stochastic gates to penalise non-zero weights during training. Variational Dropout [56] allows for unbounded dropout rates, leading to sparsity. Both are alternatives to magnitude-based pruning. Soft weight threshold reparameterisation [57] is based on learning layerwise pruning thresholds allowing non-uniform budgets across layers. Finally, the authors of [10] observe that re-training weights that previously survived pruning from their initial value can lead to sparsity at superior performance. Examples of Sparse→Sparse methods are Single Shot Network Pruning [58] where an initial mask is chosen according to the salience and kept fixed throughout. In addition, a key insight is to iteratively drop & grow weights while maintaining a fixed sparsity budget. Deep Rewiring for instance [59] augments SGD with a random walk. In addition, weights that are about to change sign are set to zero, activating other weights at random instead. Rigging the Lottery (RigL) [13] instead activates new weights by

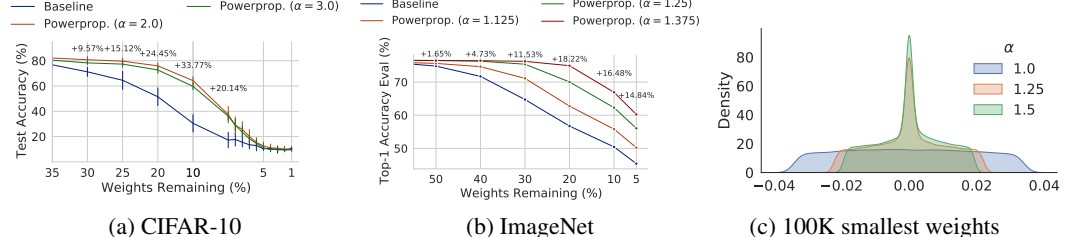

Figure 2: One-shot pruning results on CIFAR-10 (a) and ImageNet (b and c). Performance vs sparsity curves are shown in (a) and (b), highlighting significant improvements wrt. the baseline. This is due to Powerprop.'s effect on the weight distribution, which we show for the smallest 100k weights on ImageNet in (c). $\alpha > 1$ pushes the weight distribution towards zero, ensuring that more weights can be safely pruned. For CIFAR-10 we show mean and standard deviation over five runs. Repeated runs lead to almost identical results for ImageNet and are thus omitted.

highest magnitude gradients, whereas Top-KAST [14] maintains two set of masks for (sparse) forward and backward-passes. This also allows the authors to cut the computational budget for gradient computation. In addition, a weight-decay based exploration scheme is used to allow magnitude-based pruning to be used for both masks. Other notable works in this category are Sparse Evolutionary Training [12] and Dynamic sparse reparameterisation [39].

Powerpropagation is also related to Exponentiated Gradient Unnormalized [60] (Example 2) which is a reparametrisation equivalent to Powerpropagation and should be understood as a concurrent development. Indeed, the authors provide a preliminary experiment of the last layer of a convolutional network on MNIST, showing similar inherent sparsity results. We can view Powerprop. as a generalisation of this and related ideas [e.g. 19, 20, 21] with an explicit focus on modern network architectures, introducing necessary practical strategies and demonstrating the competitiveness of reparameterisation approaches at scale. In addition, the technique has not yet been used for Continual Learning.

While continual learning has been studied as early as [15, 16], the work on Elastic Weight Consolidation (EWC) [17] has recently started a resurgence of interest in the field. Contemporary approaches are subdivided into methods relying on regularisation [e.g. 44, 46, 47], data replay [e.g. 61, 62, 45, 63] or architectural approaches [e.g. 64, 65]. Works such as PackNet and SpaceNet [48, 49] have also introduced explicit gradient sparsity as a tool for more efficient continual learning. Other approaches include PathNets which [66] uses evolutionary training to find subnetworks within a larger model for individual tasks.

## 6 Experimental Evaluation

We now provide an experimental comparison of Powerpropagation to a variety of other techniques, both in the sparsity and continual learning settings. Throughout this section we will be guided by three key questions: (i) Can we provide experimental evidence for *inherent sparsity*? (ii) If so, can Powerprop. be successfully combined with existing sparsity techniques? (iii) Do improvements brought by Powerprop. translate to measurable advances in Continual Learning?

This section covers varying experimental protocols ranging from supervised image classification to generative modelling and reinforcement learning. Due to space constraints we focus primarily on the interpretation of the results and refer the interested reader to details and best hyperparameters in the Appendix.

### 6.1 Inherent Sparsity

Turning to question (i) first, let us start our investigation by comparing the performance of a pruned method without re-training. This is commonly known as the one-shot pruning setting and is thus a Dense $\rightarrow$ Sparse method. If Powerprop. does indeed lead to inherently sparse networks its improvement should show most clearly in this setting. Figure 2 shows this comparison for image classification on the popular CIFAR-10 [67] and ImageNet [68] datasets using a smaller version of

| Method | 0% | | | | |
|---|---|---|---|---|---|
| Dense | $76.8 \pm 0.09$ | | | | |
| | **80%** | **90%** | **95%** | **80% (ERK)** | **90% (ERK)** |
| Static [13] | $70.6 \pm 0.06$ | $65.8 \pm 0.04$ | $59.5 \pm 0.11$ | $72.1 \pm 0.04$ | $67.7 \pm 0.12$ |
| DSR [39] | $73.3$ | $71.6$ | | | |
| SNFS [38] | $74.2$ | $72.3$ | | $75.2 \pm 0.11$ | $72.9 \pm 0.06$ |
| SNIP [58] | $72.0 \pm 0.10$ | $67.2 \pm 0.12$ | $57.8 \pm 0.40$ | | |
| RigL [13] | $74.6 \pm 0.06$ | $72.0 \pm 0.05$ | $67.5 \pm 0.10$ | $75.1 \pm 0.05$ | $73.0 \pm 0.04$ |
| Iterative pruning [11] | $75.6$ | $73.9$ | $70.6$ | | |
| Iterative pruning (ours) | $75.3 \pm 0.07$ | $73.7 \pm 0.14$ | $70.6 \pm 0.05$ | | |
| Powerprop. + Iter. Pruning | $\mathbf{75.7} \pm 0.05$ | $\mathbf{74.4} \pm 0.02$ | $\mathbf{72.1} \pm 0.00$ | | |
| TopKAST$^\dagger$ [14] | $75.47 \pm 0.03$ | $74.65 \pm 0.03$ | $\mathbf{72.73} \pm 0.10$ | $75.71 \pm 0.06$ | $74.79 \pm 0.05$ |
| Powerprop. + TopKAST$^\dagger$ | $\mathbf{75.75} \pm 0.05$ | $\mathbf{74.74} \pm 0.04$ | $\mathbf{72.89} \pm 0.10$ | $\mathbf{75.84} \pm 0.01$ | $\mathbf{74.98} \pm 0.09$ |
| TopKAST* [14] | $76.08 \pm 0.02$ | $75.13 \pm 0.03$ | $73.19 \pm 0.02$ | $76.42 \pm 0.03$ | $75.51 \pm 0.05$ |
| Powerprop. + TopKAST* | $\mathbf{76.24} \pm 0.07$ | $\mathbf{75.23} \pm 0.02$ | $\mathbf{73.25} \pm 0.02$ | $\mathbf{76.76} \pm 0.08$ | $\mathbf{75.74} \pm 0.08$ |

Table 1: Performance of sparse ResNet-50 on Imagenet. Baseline results from [13]. ERK: Erdos-Renyi Kernel [13]. */$^\dagger$: TopKAST at 0%/50% Backward Sparsity. Shown are mean and standard deviation over 3 seeds.

AlexNet [3] and ResNet50 [4] respectively.[4] In both situations we notice a marked improvement, up to 33% for CIFAR and 18% for ImageNet at specific sparsity rates. In Fig. 2c we show the density of the smallest 100K weights, in which we observe a notable peak around zero for $\alpha > 1.0$, providing strong evidence in favour of the inherent sparsity argument. Finally is worth noting that the choice of $\alpha$ does influence the optimal learning rate schedule and best results were obtained after changes to the default schedule.

## 6.2 Advanced Pruning

| Method | 80% Sparsity | 90% Sparsity |
|---|---|---|
| Iter. Pruning $_{(1.5\times)}$ | $76.5$ [0.84x] | $75.2$ [0.76x] |
| RigL $_{(5\times)}$ | $76.6\pm_{0.06}$ [1.14x] | $75.7\pm_{0.06}$ [0.52x] |
| **ERK** | | |
| RigL $_{(5\times)}$ | $77.1\pm_{0.06}$ [2.09x] | $76.4\pm_{0.05}$ [1.23x] |
| Powerprop. + TopKAST* $_{(2\times)}$ | $77.51\pm_{0.03}$ [1.21x] | $76.94\pm_{0.10}$ [0.97x] |
| Powerprop. + TopKAST* $_{(3\times)}$ | $\mathbf{77.64}\pm_{0.05}$ [1.81x] | $\mathbf{77.16}\pm_{0.19}$ [1.46x] |

Table 2: Extended training. Numbers in square Brackets show FLOPs relative to training of a dense model for a single cycle.

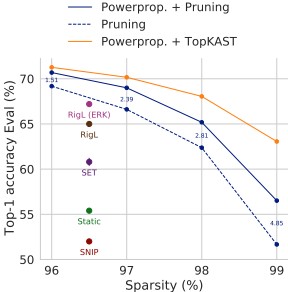

Figure 3: Extreme sparsity

We now address question (ii) by comparing Powerprop. when used in conjunction with and compared to exiting state-of-the-art algorithms on ImageNet. We orient ourselves primarily on the experimental setup in [13] & [14], both of which present techniques among the strongest in the literature. To obtain a Dense→Sparse method we combine Powerprop. with Iterative Pruning and combine TopKAST [14] with Powerprop. as our method of choice from the Sparse→Sparse literature.

We distinguish between three sub-settings a) Table 1 shows ResNet 50@$\{80\%, 90\%, 95\%\}$ sparsity, for which the largest number of baseline results are available. We also provide results using the Erdos-Renyi Kernel [13], a redistribution of layerwise sparsity subject to the same fixed overall budget. While this improves results, it increases the floating point operations (FLOPs) at test time. b) Table 2: Extended training, where we scale training steps and learning rate schedule by a factor of the usual 32K steps. While this tends to lead to better results the computational training cost increases. c) Figure 3: Extreme sparsity at $95\% - 99\%$ thus testing methods at the limit of what is currently possible. All experiments are repeated 3 times to report mean and standard deviation. As Powerprop. introduces a negligible amount of extra FLOPs (over baseline methods) we only show such values in the extended training setting to provide a fair comparison to the setup in [13]. A longer discussion of a FLOP trade-off can be found in the work on TopKAST [14].

Examining the results, we note that Powerprop. improves both Iterative Pruning & TopKAST in all settings without any modification to the operation of those algorithms. In all settings a large array

---

[4]The motivating example in Section 3 also follows this protocol.

of other techniques are outperformed. To the best of our knowledge those results constitute a new state-of-the art performance in all three settings. We emphasise that many of the methods we compare against are based on pruning weights by their magnitude and would thus also be likely to benefit from our method.

## 6.3 Continual Learning

### 6.3.1 Overcoming Catastrophic forgetting

| Algorithm | MNIST | notMNIST |
|---|---|---|
| Naive | $-258.40$ | $-797.88$ |
| Laplace [69, 65] | $-108.06$ | $-225.54$ |
| EWC [17] | $-105.78$ | $-212.62$ |
| SI [44] | $-111.47$ | $-190.48$ |
| VCL [46] | $-94.27$ | $-187.34$ |
| Eff. PackNet (EPN) | $-94.50$ | $-177.08$ |
| Powerprop. + Laplace | -96.72 | -196.87 |
| Powerprop. + EWC | -95.79 | -188.56 |
| Powerprop. + EPN | **-92.15** | **-174.54** |

Table 3: Results on the Continual generative modelling experiments. Shown is an importance-sampling estimate of the test log-likelihood.

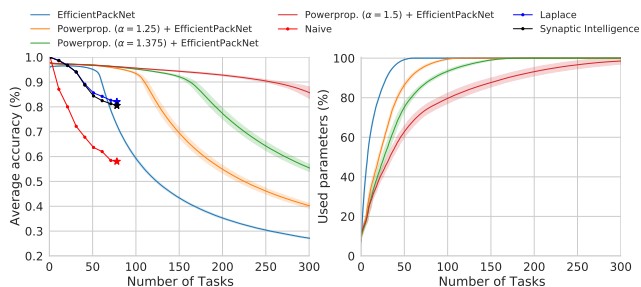

Figure 4: Powerpropagation on extremely long task sequences. **Left**: Average accuracy over all tasks. **Right**: Fraction of unmasked parameters. Shown are mean and standard deviation over 10 repetitions of the experiment.

We now turn to question (iii), studying the effectiveness of sparsity methods for Continual Learning. We first consider long task sequences on the Permuted MNIST benchmark, first popularised by [70]. While usually regarded to be an idealised and simple setting, we push its boundaries by drastically increasing the number of tasks, which was previously limited to 100 in [71] and 78 in [72], whereas we learn up to 300 in a single model. In all cases we use a network with a single output layer across all tasks, following the protocol by and comparing to [72]. Figure 4 shows the average accuracy and fraction of used weights. We significantly outperform baselines with an average acc. of $\approx 86\%$ after 300 tasks, whereas the closest baseline (Laplace [65]) falls to 85.61% after merely 51 tasks. This also highlight the effect of $\alpha$ more noticeably, showing increased performance for higher $\alpha$. Note also that the gradient of the used weights curve flattens, showcasing the dynamic allocation of larger number of parameters to tasks early in the sequence, whereas later tasks reuse a larger fraction of previous weights. While we also attempted a comparison with standard PackNet, the model failed, as its default allocation of <0.4% of weights per task is insufficient for even a single MNIST task (see Figure 1a). The perspective reader might notice higher performance for considered baselines at small number of tasks. This is due to our choice of $\gamma = 0.9$ (see Section 4) which we optimise for long sequences. It is worth mentioning that $\alpha$ can be increased up to a certain limit at which point training for high number of tasks can become unstable. We will investigate this effect in future work.

Moving onto more complex settings, we use Powerprop. on the continual generative modelling experiment in [46] where 10 tasks are created by modelling a single character (using the MNIST and notMNIST[5] datasets) with a variational autoencoder [73]. We report quantitative results in Table 3. In both cases we observe Powerprop. + EfficientPackNet outperforming the baselines. Interestingly, its effectiveness for Catastrophic Forgetting is not limited to PackNet. In particular, we provide results for the Elastic Weight Consolidation (EWC) method and its online version (Laplace) [17, 65] both of which are based on a Laplace approximation (using the diagonal Fisher information matrix). As Powerprop. results in inherently sparse networks, the entries of the diagonal Fisher corresponding to low magnitude weights are vanishingly small (i.e. the posterior distribution has high variance), resulting in virtually no regularisation of those parameters. We observe significant improvements with no other change to the algorithms.

Finally, we move onto Catastrophic Forgetting in Reinforcement Learning using six tasks from the recently published Continual World benchmark [74], a diverse set of realistic robotic manipulation tasks. This is arguably the most complex setting in our Continual Learning evaluation. Following the authors, we use Soft actor-critic [75], training for 1M steps with Adam [33] (relying on the formulation in Section 2) on each task while allowing 100k retrain steps for PackNet (acquiring no

---

[5]http://yaroslavvb.blogspot.com/2011/09/notmnist-dataset.html

Table 4: Reinforcement Learning Results on Continual World [74]. Error bars provide 90% confidence intervals.

| Method | Avg. success | Forgetting |
|---|---|---|
| Fine-tuning | 0.00 [0.00, 0.00] | 0.87 [0.84, 0.89] |
| A-GEM [62] | 0.02 [0.01, 0.04] | 0.89 [0.86, 0.91] |
| EWC [17] | 0.66 [0.61, 0.71] | 0.07 [0.04, 0.11] |
| MAS [78] | 0.61 [0.56, 0.66] | -0.01 [-0.04, 0.01] |
| Perfect Memory | 0.36 [0.32, 0.40] | 0.07 [0.05, 0.10] |
| VCL [46] | 0.52 [0.47, 0.57] | -0.01 [-0.03, 0.01] |
| PackNet [48] | 0.80 [0.76, 0.84] | 0.02 [0.04, 0.00] |
| Power. + EfficientPackNet $_{(\alpha=1.375)}$ | 0.82 [0.77, 0.87] | 0.01 [0.03, 0.01] |
| Power. + EfficientPackNet $_{(\alpha=1.5)}$ | **0.86** [0.82, 0.90] | **0.00** [-0.02, 0.02] |

additional data during retraining). We report the average success rate (a binary measure based on the distance to the goal) and a forgetting score measuring the difference of current performance to that obtained at the end of training (but importantly before pruning) on a certain task. Thus, if Powerprop. does improve results, this will show in the Forgetting metric. Results are shown in Table 4. Indeed, we notice that a higher $\alpha$ leads to a reduction of forgetting and hence an overall improvement over PackNet, resulting in superior performance. Finally, it is worth mentioning that (Efficient)PackNet still assume hard task boundaries that are known during training. However, it has previously been shown that it is possible to overcome these limitations using a changepoint detection algorithm [e.g. 76, 77].

## 7    Discussion

In this work we introduced Powerpropagation, demonstrating how its effect on gradients leads to *inherently sparse* networks. As we show in our empirical analyses, Powerprop. is easily combined with existing techniques, often merely requiring a few characters of code to implement. While we show its effect on only a few established algorithms, we hypothesise that a combination with other techniques could lead to further improvements. Bridging gaps between the fields of sparse networks and Continual Learning, we hope that our demonstration of its effect on Catastrophic Forgetting will inspire further interest in such ideas. In terms of future work, we believe that devising automated schemes for learning $\alpha$ may be worthwhile, making the use of our method even simpler. Particular attention ought to be paid to its interaction with the learning rate schedule, which we found worthwhile optimising for particular choices of $\alpha$. Finally, we also hope further work will bring additional theoretical insights into our method.

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
