## A  Funding Transparency Statement

The authors received no specific funding for this work.

## B  Powerpropagation with modern optimisers

Figure 5 shows an empirical evaluation on MNIST relating to our discussion of modern optimisation algorithms in Section 2. We compare Adam [33] (with a learning rate of $10^{-3}$) applied to Powerpropagation a) directly to compute $\frac{d\mathcal{L}}{dW}$ and our proposed version b) $Adam(\frac{d\mathcal{L}}{dW^*}) \cdot \frac{dW^*}{dW}$ where we denote by $W^*$ the re-parameterised version of $W$. For completeness, we also include a standard classifier not using Powerpropagtion trained with Adam.

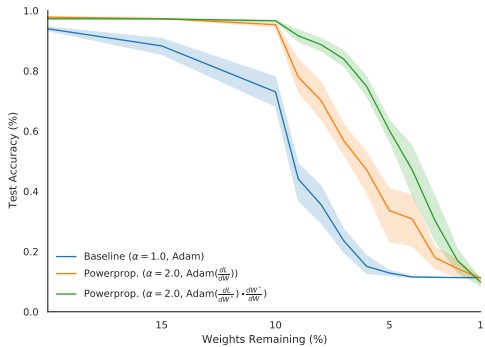

Figure 5: Shown are mean and standard deviation over five random seeds.

We note that performance of our alternative gradient computation indeed performs better than vanilla Adam. Nevertheless, its distinct improvement over the baseline is notable, suggesting that the issue pointed out in Section 2 may be less severe than expected. We hypothesise this is due to the moving average computation of Adam, suggesting Adam is unable to completely undo the "rich get richer" dynamics introduced by Powerpropagation, underestimating the correction at every point. This suggest that Adam's parameters ($\beta_1$, $\beta_2$) may play an important role. We lave this investigation for future work and recommend our computation $Adam(\frac{d\mathcal{L}}{dW^*}) \cdot \frac{dW^*}{dW}$ as a default choice. While the result above is merely presented on the simple MNIST dataset, we observed similar behaviour on other experiments presented.

## C  Task inference during Evaluation

A possible concern for the method presented for Continual Learning is the requirement of known task ids at inference time, which are needed to select the correct mask for inputs from an unknown (but previously encountered) problem. This has recently been identified to be a key consideration, distinguishing between the Continual Learning sub-cases of Task-incremental (For classification: With task given, which label?) and Class-incremental learning (Which task and which label?) [72, 79]. Importantly, several methods shown to perform well in the first setting show significant shortcomings when task-ids are not provided.

To overcome this, we propose the use of an unsupervised mask-inference technique recently proposed in [80]: At its core it makes use of a mixture model with each component defined by the subnetwork for task $t$ (corresponding to mask $\mathcal{M}^t$) and mixture coefficients given by a $T$-dimensional vector $\pi$:

$$\mathbf{p}(\pi) = f\left(\mathbf{x}, \phi \odot \left(\sum_{t=1}^{T} \pi_t \mathcal{M}_t\right)\right) \tag{2}$$

where $\mathbf{x}$ is the input and $f$ an appropriate function that returns a predictive density (e.g. a softmax). In the absence of an output or task-id associated with $\mathbf{x}$, the authors propose the minimisation of the entropy $\mathcal{H}(\mathbf{p}(\pi))$ as an objective to identify the correct mask. The intuition is that a sub-network operating in the space of the input corresponding to $\mathbf{x}$'s task-id should have low entropy. Thus, the

negative gradient $-\frac{\partial \mathcal{H}(\mathbf{p}(\pi))}{\partial \pi}$ with respect to the mixture coefficients provides useful information. In its more advanced version the procedure then eliminates half of the mixture coefficients (with lowest gradient value) at each step, constructs a new mixture out of the remaining coefficients and repeats the procedure until a single mask remains. The task-id is thus estimated in logarithmic time. This is shown to work well even for thousand of tasks and can be readily combined with Efficient PackNet and Powerpropagation.

It is worth mentioning that the assumption of adequately calibrated predictive confidence in standard neural networks is possibly a naive one given their well-known overconfidence outside the training distribution [e.g. 81]. However, the generality of powerpropagation suggests that it might be readily combined with techniques designed to improve on calibrated uncertainty [e.g. 82, 83] and thus make the suggested inference procedure more robust.

We now test the task-inference procedure detailed in Section C in two settings: (1) The class-incremental learning versions of the Permuted- and Split-MNIST benchmark which we take from [72] and (2) The more challenging 10-task Split-CIFAR100 benchmark using an AlexNet-style ConvNet. Note that the protocols between the two benchmark vary slightly, as (2) allows for separate output-layers/heads whereas (1) insists on a single such head. Also, baseline results in (2) do not strictly follow the class-incremental learning setting but are included due to being among the most recent and well-performing results on the benchmark. The choice of these datasets allow us to compare Powerpropagation and EfficientPackNet together with the task-inference procedure against a plethora of published models. In all cases we chose hyperparameters on the validation set first before obtaining test set performance.

Table 5: Class-incremental learning results on Permuted- and Split-MNIST and 10-task Split-CIFAR 100. MNIST baseline results are taken from [72], Split-CIFAR100 results from [84]. Shown are mean and standard deviation over 10 runs for MNIST and 5 runs for CIFAR100.

| Algorithm | Permuted-MNIST | Split-MNIST | 10-task Split-CIFAR100 |
|---|---|---|---|
| *Class-incremental learning* | | | |
| Naive (SGD) | 12.82%±0.95 | 19.46%±0.04% | |
| EWC [17] | 26.32%±4.32 | 19.80%±0.05% | |
| Laplace [65] | 42.58%±6.50 | 19.77%±0.04% | |
| SI [44] | 58.52%±4.20 | 19.67%±0.09% | |
| MAS [78] | 50.81%±2.92 | 19.52%±0.29% | |
| LwF [85] | 22.64%±0.23 | 24.17%±1.33% | |
| EfficientPackNet | 97.09%±0.08 | 99.42%±0.15% | |
| GEM [86] | 96.19%±0.11 | 96.16%±0.35% | |
| DGR [61] | 95.09%±0.04 | 95.74%±0.23% | |
| RtF [87] | 97.06%±0.02 | 97.31%±0.11% | |
| Powerprop. + EfficientPackNet | **97.47% ± 0.05** | **99.71% ± 0.04%** | |
| *Separate heads + Given task-ids* | | | |
| OWM [88] | | | 50.94% ± 0.60 |
| A-GEM [62] | | | 63.98% ± 1.22 |
| EWC [17] | | | 68.80% ± 0.88 |
| ER_Res [45] | | | 71.73 ± 0.63% |
| HAT [89] | | | 72.06% ± 0.50 |
| GPM [84] | | | 72.48 ± 0.40% |
| *Separate heads + Task inference* | | | |
| Powerprop. + EfficientPackNet | | | **73.70 ± 0.70** |

We note that Powerpropagation + PackNet performs superior in all settings, suggesting that the task-inference procedure works well at least for medium-length task-sequences. To the best of our knowledge, the 10-task Split-Cifar100 results constitute a new state of the art. Hyperparameters for all Continual Learning benchmarks are shown in the Appendix.

## D  Inherent sparsity/One-shot pruning

We show hyperparameters for for MNIST and CIFAR-10 in Table 6. The MNIST values correspond to the motivating experiment in Section 3. For clarity, we include the one-shot experiment on ImageNet along with the more advanced sparsity techniques in Table 7. Our selection of hyperparameters was based on final performance of the dense model.

## E  Advanced Pruning

Our implementation of TopKAST is based on code kindly provided to us by the authors, thus ensuring reproducibility of published results. While we were unable to obtain a reference implementation for Iter. Pruning, we do include a comparison of our implementation to results published in [11]. Our results match published results closely. We suggest the small gap may be closed by a more extensive sweep over update intervals and the number of training iterations upon which pruning starts or stops respectively. For our experiments on ImgageNet the treatment of the first and final layer in the network varies between different experimental protocols. As we apply Powerprop. in conjunction with various techniques we follow the protocol outlined below:

- **Iter. Pruning, Powerprop. + Iter. Pruning** We follow the protocol in [11] *"We modify our ResNet-50 training setup to leave the first convolutional layer fully dense, and only prune the final fully-connected layer to 80% sparsity. This heuristic is reasonable for ResNet-50, as the first layer makes up a small fraction of the total parameters in the model and the final layer makes up only .03% of the total FLOPs."*
- **TopKAST, Powerprop. + TopKAST** We follow the protocol in [14] using the code supplied by the authors. Neither the final fully-connected layer nor the first layer are pruned.
- **Powerprop. + TopKast + ERK** We follow the ERK sparsity distribution for 80% & 90% global sparsity shown in [13] (Figure 12), which prunes both the first and final layer according to the ERK distribution.

Furthermore, it is worth mentioning that it is customary to initialise the final (dense) layer to zero, which would make learning with Powerpropagation impossible due to the critical point property discussed in Section 2. Thus, we choose the initialiser in accordance with all other convolutional layers.

We show hyperparameters for our Imagenet experiments in Table 7. Note that we follow the learning rate schedule introduced by [90], where the learning rate is $\eta = \eta_b 0.1 \cdot \frac{kn}{256}$ where $\eta_b$ is the base learning rate (0.1 by default) and $kn$ is the minibatch size.

## F  Continual Learning on Deep Generative Models

Figures 6 & 7 show full results for the Continual Learning experiment with Deep Generative Models introduced by [46]. Note that catastrophic forgetting is eliminated by construction for approaches involving PackNet (due to the masking of gradients), explaining the improved performance. We also show a summary of the hyperparameters required to replicate the results in Table 8. We choose the best hyperparameters based on results on the validation set before computing final test-set results (using 5,000 MC samples to approximate the log-likelihood) using the best values previously chosen.

Our method was built on top of the published code (`https://github.com/nvcuong/variational-continual-learning`). The MNIST [41] dataset was downloaded from `http://yann.lecun.com/exdb/mnist/`, the notMNIST dataset is available here: `http://yaroslavvb.blogspot.com/2011/09/notmnist-dataset.html`.

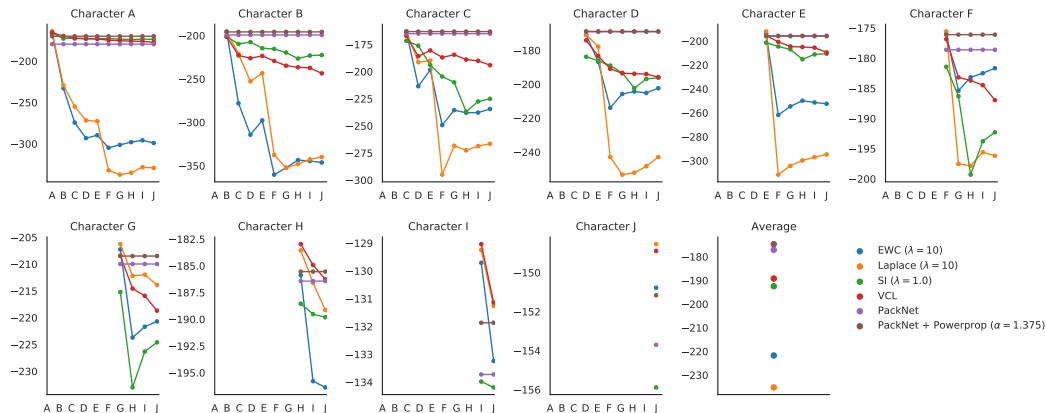

Figure 7: Test-LL results on notMNIST. The higher the better.

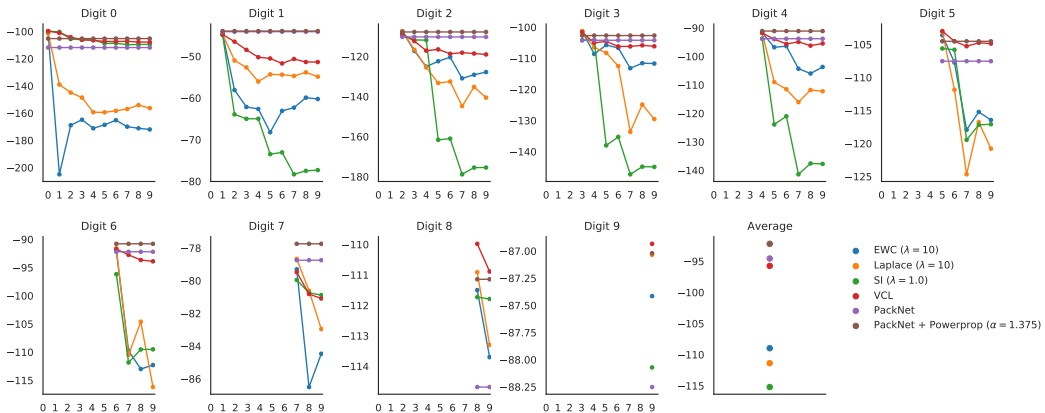

Figure 6: Test-LL results on MNIST. The higher the better.

## G Continual Reinforcement Learning

Our experiments on Reinforcement Learning follow the protocol in [74]. We built Powerprop. + Eff. PackNet on top of the authors' code (`https://github.com/zajaczajac/cl-benchmark-tf2`). Note that due to computational constraints, we merely show results on six out of ten of the original CW10 task sequence. The full is of tasks is (in order of training):

1) 'hammer-v1'
2) 'push-wall-v1',
3) 'faucet-close-v1'
4) 'push-back-v1'
5) 'stick-pull-v1'
6) 'handle-press-side-v1'

A full description of each task is available in [91]. Hyperparameters are shown in Table 9 and were chosen based on the best results on the original three tasks, before running the final version on the full sequence. Note that we keep the vast majority of training settings unchanged. As the sparsity search procedure in Section 4 would include further interaction with the environment, we disable it for a fair comparison and reserve a fixed sparsity of 10% per task.

# H   Task inference

Hyperparameters for the Class-Incremental learning versions of Permuted- and Split-MNIST are shown in Table 10 and for 10-task Split-Cfiar100 in Table 11.

# I   Hyperparameters

Table 6: Hyperparameters for the pruning experiments on MNIST & CIFAR-10.

| Parameter | Considered range | Comment |
|---|---|---|
| **All Datasets** | | |
| Activation function | $\{f(x) : \max(0, x) \text{ (ReLu)}\}$ | [10] |
| Batch size | $\{60\}$ | " |
| Pruning | {Layerwise} | " |
| Sparsity at output layer | $\{0.5\times\}$ | " |
| Initialisation | {Gaussian Glorot} | " |
| Optimiser | {Momentum (0.9)} | |
| Learning rate | $\{\mathbf{0.0025}\}$ | |
| Powerprop. $\alpha$ | $\{2, 3, 4, 5\}$ | Unstable for $\alpha > 5$ |
| **MNIST** specific | | |
| Network | $\{[784, 300, 100, 10]\}$ | (LeNet) in [10] |
| Training steps | $\{50000\}$ | |
| **CIFAR-10** specific | | |
| Convolutions | $\{[64 \times 2, pool, 128 \times 2, pool, 256 \times 2, pool]\}$ | (Conv-6) in [10] |
| Fully Connected Layer | $\{[256, 256, 10]\}$ | " |
| Conv. Filters | $\{[3 \times 3]\}$ | " |
| *pool* | {[max-pooling (stride 2)]} | " |
| Training steps | $\{100000\}$ | |
| Prune Batchnorm variables | {False} | " |

Table 7: Hyperparameters for the experiments on ImageNet.

| Parameter | Considered range | Comment |
|---|---|---|
| Network | ResNet-50 | [4] |
| Batch size | {4096} | [14] |
| Training steps | {32000} | " |
| Learning rate ramp up epochs | {5} | " |
| Layerwise pruning? | {True} | |
| Weight decay | {$10^{-4}$} | " |
| Label smoothing | {0.1} | " |
| Learning rate drops ($\times 0.1$) after epochs | {[30, 70, 90]} | " |
| Initialise last year to zero? (All other methods) | {True} | |
| Initialise last year to zero? (TopKAST, Powerprop.) | {False} | |
| Optimiser | {Nesterov Momentum (0.9)} | |
| Batchnorm decay rate | {0.9} | |
| Prune Batchnorm variables | {False} | " |
| **Powerprop. + One-shot Pruning** specific | | |
| Powerprop. $\alpha$ | {1.125, 1.25, **1.375**} | |
| Base Learning rate ($\eta_b$) | {0.1, 0.25, **0.5**} | |
| Prune Logits | {True} | " |
| Prune Initial Conv. | {False} | " |
| **Powerprop. + Iter. Pruning** specific | | |
| Base Learning rate ($\eta_b$) | {0.1, **0.4**, 0.5, 0.6} | |
| Start pruning after n% of training | {0, 10, **20**, 30} | |
| Stop pruning after n% of training | {70, **80**, 90, 100} | |
| Pruning interval | {2000} | |
| Powerprop. $\alpha$ | {1.125, 1.25, **1.375**, 1.5, 2.0} | |
| **Powerprop. + TopKAST** specific | | |
| Base Learning rate ($\eta_b$) | {**0.1**, 0.4} | |
| Start pruning after n% of training | {0, 10, **20**, 30} | |
| Stop pruning after n% of training | {70, **80**, 90, 100} | |
| Pruning interval | {2000} | |
| Powerprop. $\alpha$ | {1.125, 1.25, **1.375**, 1.5, 2.0} | |

Table 8: Hyperparameters for the experiments on Deep Generative Models.

| Parameter | Considered range | Comment |
|---|---|---|
| **Both Datasets** | | |
| Network | $\{[784, 500, 500, 500, 100]\}$ | [72] |
| Task specific decoder | $\{[50, 500, 500]\}$ | " |
| Shared decoder | $\{[500, 500, 784]\}$ | " |
| Activation function | $\{f(x) : \max(0, x) \text{ (ReLu)}\}$ | " |
| MC-Samples at Evaluation (All Methods) | $\{5000\}$ | " |
| Batch size | $\{50\}$ | " |
| $\dim(z)$ | $\{50\}$ | " |
| Regularisation strength ($\lambda$) for EWC, Laplace | $\{1, \mathbf{10}, 100\}$ | " |
| Regularisation strength ($\lambda$) for SI | $\{\mathbf{1}, 10, 100\}$ | " |
| MC-Samples during Training (for VCL) | $\{10\}$ | " |
| Freeze biases after first task? | $\{\text{True}\}$ | " |
| $\gamma$ for EfficientPackNet? | $\{0.95\}$ | " |
| Optimiser for VCL, SI, EWC, Laplace | Adam | " |
| Learning rate for Adam | $\{10^{-3}\}$ | " |
| Optimiser (PackNet, All Powerprop. methods) | $\{\text{Momentum (0.9)}\}$ | |
| **MNIST** specific | | |
| Learning rate for Momentum | $\{[\mathbf{1}, 2.5, 5, 7.5] \cdot 10^{-3}, 10^{-2}\}$ | |
| $\alpha$ for Powerprop. | $\{1.125, 1.25, 1.375, 1.5, 2.0\}$ | See Table for best value. |
| Training epochs (SI) | $\{400\}$ | " |
| Training epochs (all others) | $\{200\}$ | " |
| **notMNIST** specific | | |
| Learning rate for Momentum | $\{[1, \mathbf{2.5}, 5, 7.5] \cdot 10^{-3}, 10^{-2}\}$ | |
| $\alpha$ for Powerprop. | $\{1.125, 1.25, 1.375, 1.5, 2.0\}$ | See Table for best value. |
| Training epochs (all) | $\{400\}$ | " |

Table 9: Hyperparameters for the experiments on Continual Reinforcement Learning.

| Parameter | Considered range | Comment |
|---|---|---|
| Network | $\{[12, 256, 256, 256, 256]\}$ | [74] |
| Actor output | $\{4\}$ | " |
| Critic output | $\{1\}$ | " |
| Activation | $\{\text{Leaky ReLU}\}$ | " |
| Task-specific heads | $\{\text{True}\}$ | " (Actor only) |
| Layernorm | $\{\text{True}\}$ | " |
| Regularise critic | $\{\text{False}\}$ | " |
| Reset Critic on Task Change | $\{\text{True}\}$ | " |
| Batch size | $\{128\}$ | " |
| Discount factor | $\{0.99\}$ | " |
| Target output std. $\sigma_t$ | $\{0.089\}$ | " |
| Replay buffer size | $\{1\text{M per task}\}$ | " |
| Replay buffer type | $\{\text{FIFO}\}$ | " |
| Start training after | $\{1\text{k steps}\}$ | " (for each task) |
| Train every | $\{50 \text{ steps}\}$ | " |
| Update steps | $\{50 \text{ steps}\}$ | " (i.e. 50 updates every 50 training steps) |
| Training steps | $\{1\text{m/task}\}$ | " |
| Eff. PackNet re-train steps | $\{100\text{k/task}\}$ | " (Using exiting replay data.) |
| Optimiser | $\{\text{Adam}\}$ | " |
| Learning Rate | $\{10^{-3}\}$ | " |
| Reset Optimiser on Task Change | $\{\text{True}\}$ | " |
| Gradient norm clipping | $\{2 \cdot 10^{-5}, \mathbf{10^{-3}}\}$ | |
| Powerprop. + Eff. PackNet specific | | |
| Layerwise pruning? | $\{\mathbf{False}, \text{True}\}$ | |
| Sparsity range $S$ | $\{[0.1]\}$ | |
| Powerprop. $\alpha$ | $\{1.125, 1.25, 1.375, \mathbf{1.5}\}$ | |
| Re-initialise pruned weights? | $\{\text{True}\}$ | |

Table 10: Hyperparameters for the experiments on Permuted- & Split-MNIST. EPN: Efficient PackNet

| Parameter | Considered range | Comment |
|---|---|---|
| **All Datasets** | | |
| Activation function | $\{f(x) : \max(0, x) \text{ (ReLu)}\}$ | [72] |
| Task-specific heads? | {True} | " |
| Batch size | {64} | |
| Pruning | {Layerwise, **Full Model**} | |
| Optimiser | {SGD} | |
| Sparsity range $S$ | $\{[0.9, \ldots 0.3, 0.25 \ldots 0.15, 0.14, \ldots 0.01]\}$ | |
| ] Convolutions | $\{[64 \times 2, pool, 128 \times 2, pool, 256 \times 2, pool]\}$ | (Conv-6) in [10] |
| Fully Connected Layer | $\{[256, 256, 10]\}$ | " |
| Conv. Filters | $\{[3 \times 3]\}$ | " |
| *pool* | {[max-pooling (stride 2)]} | " |
| Training steps | {100000} | |
| Prune Batchnorm variables | {False} | " |
| Network | $\{[784, 1000, 1000, 10]\}$ | [72] |
| Re-initialise pruned weights? | {True} | |
| Target performance factor $\gamma$ for EPN | $\{0.9, 0.95, \mathbf{0.99}, 0.995\}$ | |
| $\gamma$ for Powerprop. + EPN | $\{0.9, 0.95, \mathbf{0.99}, 0.995\}$ | |
| Powerprop. $\alpha$ | $\{1.125, 1.25, \mathbf{1.375}, 1.5, 2.0\}$ | |
| Training steps per Task | {50000} | |
| PackNet retrain steps | $\{0, \mathbf{50000}\}$ | |
| Learning rate | $\{0.1, \mathbf{0.05}, 0.01\}$ | |
| Network | $\{[784, 1000, 1000, 10]\}$ | [72] |
| Re-initialise pruned weights? | {True} | |
| Target performance factor $\gamma$ for EPN | $\{\mathbf{0.9}, 0.95, 0.99, 0.995\}$ | |
| $\gamma$ for Powerprop. + EPN | $\{\mathbf{0.9}, 0.95, 0.99, 0.995\}$ | |
| Powerprop. $\alpha$ | $\{1.25, 1.375, \mathbf{1.5}, 2.0\}$ | $\alpha \geq 2.0$ unstable |
| Training steps per Task | {50000} | |
| PackNet retrain steps | {50000} | |
| Learning rate | $\{0.1, \mathbf{0.05}, 0.01\}$ | |

Table 11: Hyperparameters for the experiments on 10-task Split-Cifar100. EPN: Efficient PackNet

| Parameter | Considered range | Comment |
|---|---|---|
| **All Datasets** | | |
| Activation function | $\{f(x) : \max(0, x) \text{ (ReLu)}\}$ | [84] |
| Task-specific heads? | {True} | " |
| Batch size | {64} | " |
| Pruning | {Layerwise, **Full Model**} | |
| Optimiser | {SGD} | " |
| Sparsity range $S$ | $\{[0.9, \ldots 0.3, 0.25 \ldots 0.15, 0.14, \ldots 0.01]\}$ | |
| ] Convolutions | $\{[64, pool, 128, pool, 256, pool]\}$ | " |
| Fully Connected Layer | $\{[2048, 2048]\}$ | " |
| Conv. Filters | $\{[4 \times 4, 3 \times 3, 2 \times 2]\}$ | " |
| *pool* | {[max-pooling (stride 2)]} | " |
| Batchnorm | {False} | " |
| Dropout | $\{0.2, 0.2, 0.5, 0.5, 0.5, 0.0\}$ | Specifying rates per layer |
| Re-initialise pruned weights? | {True} | |
| $\gamma$ for Powerprop. + EPN | $\{0.9, 0.95, \mathbf{0.99}, 0.995\}$ | |
| Powerprop. $\alpha$ | $\{1.375, 1.5, 1.625, \mathbf{1.75}\}$ | |
| Training steps per task | $\{50000, \mathbf{75000}\}$ | |
| PackNet retrain steps | $\{0, \mathbf{\text{Training steps}}\}$ | |
| Learning rate | $\{0.1, \mathbf{0.05}\}$ | |