# OpenReview forum: "Powerpropagation: A sparsity inducing weight reparameterisation"
_NeurIPS.cc/2021/Conference — NeurIPS 2021 Poster_

### Official Review · Reviewer_3XMd · 2021-06-30

**Rating:** 8
**Confidence:** 5

**Summary:**

This is another paper where the goal and idea of the paper are almost summarized well in the title and the abstract. The only missing part is the parameterization itself which is very simple yet effective. A weight, w, is reparameterized using another underlying variable, v, as w = v|v|^($\alpha$ - 1), essentially reparameterizing the weight (real number) using an exponentiation operation as the name suggests.

This decoupling as the paper discusses in L106-115 leads to sparse solutions, due to the gradient scaling of the underlying variable resulting in the rich getting richer due to the exponentiation involved. The paper uses this observation, coupled with some toy understanding on MNIST gave the motivation to use this reparameterization to naturally induce sparser solutions.

The paper focuses on coupling this with sparsity-inducing methods and continual learning methods that use sparsity to encode multiple sub-networks and shows that combining this reparameterization with the SOTA techniques further improves accuracy with minimal overhead.

**Limitations And Societal Impact:**

Limitations are decently discussed and the authors seem to be aware of the potential of the method.

**Main Review:**

Let me start with a general comment before going into more details.

The paper is very engaging to read but sometimes becomes dense because of the things and ideas moving around. I liked it very much once I understood what the authors were trying to do, but long bodies of text couple with notations at places made it harder at times. It is a simple idea and the explanation could be made simpler.

Especially Section 2, it very strong but at times tries to explain everything at a stretch creating a chance to miss things. It would be great to have a simple running example with a gradient update equation would have been great like. Instead of update being w - $\eta\Delta$w the update now becomes something like w = sign(v)(v - $\eta\Delta$v)^$(\alpha)$ = sign(v)(v - $\eta\Delta$w$\alpha$|v|^$(\alpha-1)$)^$(\alpha)$ would have made things clearer. Even now there are two cases when v > 1 and v < 1 due to the nature of exponentiation, making gradient scaling while monotonic, not trivial to understand. Most weights are < 1 so the dynamics of a weight with $\alpha=2$ and v = 0.1 vs v = 0.2 do differ when looked at from w's perspective.

I strongly hope either author comment on this or fix this in the next revisions making it easier to follow.

I think that is my only major concern in the whole paper. Now coming to the strengths.

The authors identified extremely appropriate applications and showed statistically significant improvement when using powerpropagation instead of the normal weights. Both the sub-network-based CL and sparse network learning benefit from this simple yet powerful idea.

The experiments are very thorough and the authors should be appreciated for their efforts.

**Time Spent Reviewing:**

4 hrs

---

> ### Author Response · Authors · 2021-08-09
> **Response to reviewer 3XMd**
>
> As some of the comments and questions by other reviewers show, the reviewer is completely correct and justified in suggesting that the presentation of the ideas (especially in Section 2) are deserving of more space and perhaps an intuitive example. As an author, it can often be difficult to asses how easily one's own ideas are understood by someone unfamiliar with the project. We will thus do our best in improving this section and would like to thank the reviewer for this valuable comment that will help us improve the manuscript.

---

> > ### Comment · Reviewer_3XMd · 2021-08-14
> > **Thanks for the response. Will get back.**
> >
> > Dear Authors,
> >
> > Thanks for the response. I am traveling right now and will try to get back on this in a week or so. Apologies for the delay.

---

> > ### Comment · Reviewer_3XMd · 2021-08-15
> > **Voting for acceptance**
> >
> > Dear Authors,
> >
> > Thanks for acknowledging the issues in the presentation. I hope to see these changes in the camera ready.
> >
> > All the best.

---

### Official Review · Reviewer_m5ux · 2021-07-17

**Rating:** 7
**Confidence:** 3

**Summary:**

This paper is focused on weight reparameterization, proposing a versatile method that leads to inherently sparse models.  The authors theoretically analyze this extension and backup the effectiveness of their proposed method with encouraging empirical results.

**Limitations And Societal Impact:**

Yes.

**Main Review:**

The basic idea of altering the gradient descent behavior by an approximated factor, exhibiting a "rich get richer" dynamic, is well-worthed and original.  Several experiments are conducted to demonstrate the performance of the proposed model. Promising results in terms of accuracy have been achieved in combining the proposed method -- Powerpropagation, with interactive pruning methods and sparse-to-sparse algorithms with a slight increase in FLOPs. Still, it is not clear what is the impact of alpha. Can you provide a more systematic analysis?

I like the diversity and the choice of the type of experiments performed and the special attention accorded to the catastrophic forgetting phenomenon in continual learning. However, the bridge between sparse training networks and continual learning appears possible in previous works (e.g. [1]). Please can you comment on the expected impact of SpaceNet on Figure 4?  Perhaps this shall be reflected in some nuances added to the discussion (rows 342-344).

[1] G. Sokar et al. "SpaceNet: Make Free Space For Continual Learning", https://arxiv.org/abs/2007.07617, 2020

**Time Spent Reviewing:**

4h

---

> ### Author Response · Authors · 2021-08-09
> **Response to reviewer m5ux**
>
> We thank the reviewer for their time and effort in providing feedback for this work.
>
> More systematic analysis of the effect of alpha: The best way to summarise our current comment on the effect of alpha is that the "rich get richer" dynamic becomes more pronounced with higher values of alpha, which ought to be increased to a point where current hyperparameter settings (learning rate etc.) are no longer sufficient to match performance of lower values. Thus, training the network at significantly higher values may require a more extensive hyperparameter search. However, a significant improvement can already be achieved by lower values of alpha as the magnitude of the weight already appears in the update.
>
> SpaceNet: Thanks for making us aware of this work, which is indeed closely related to our research. One of the ideas that distinguishes SpaceNet from PackNet (and EPN) is that sparse connections are allocated as soon as a new task is encountered. This makes it similar in spirit to the TopKAST method. A priori, our expectation is that this should work equally well or better and may be relatively easily combined with Powerpropagation (as we show TopKAST can be used with Powerprop.). We will try aim to implement Powerprop. with SpaceNet and show results in a camera-ready version. We are optimistic about expected results.

---

### Official Review · Reviewer_kFtF · 2021-07-20

**Rating:** 7
**Confidence:** 4

**Summary:**

The paper proposes a method to parameterize weights such that the resulting model is inherently sparse. This is achieved by dynamically updating the parameters proportional to their magnitude i.e., the higher magnitude weights receive larger updates while the low magnitude weights remain unaffected. The idea is novel and useful for training sparse networks geared towards applications in resource-constrained environments. Additionally, as elaborated in the paper, it is also useful in the cases of continual learning where a single model is used to learn multiple tasks. The claims are supported by experimental evaluation.

**Limitations And Societal Impact:**

Limitations are discussed. No adverse societal impact.

**Main Review:**

Strengths

- The idea of transforming the parameters with an exponent is a simple yet effective technique.
- The paper is well written with necessary mathematical derivations. Experiments adequately substantiate the proposed method and claims.
- The perspective of using sparsity in learning better models in continual learning is interesting and is very useful.

Weaknesses

- Section 3 last paragraph: The authors mention the concern that powerpropagation can lead to identical masks at initialization and at convergence which is intuitive as the small weights are not modified frequently. But it is also mentioned that the empirical results indicate that this is not the case. Can you please provide a possible justification?
- Can you please discuss how the exponentiation might impact the efficiency of the network in resource-constrained devices as exponentiation can be an expensive operation. This can increase the training complexity multiple folds.
- Section 6.3: Please mention the model architecture used for training the permuted MNIST dataset. The results in this section are impressive, however, it is limited to only one dataset in each of standard image classification and reinforcement learning settings. Moreover, permuted MNIST is synthetically generated. Hence, including another continual learning benchmark like CIFAR100 split into 10 or 20 tasks will add more value to the paper.

Overall a good paper. Minor improvements will help add more value to the paper.

**Time Spent Reviewing:**

4

---

> ### Author Response · Authors · 2021-08-09
> **Response to reviewer kFtF**
>
> We thank the reviewer for their time and effort in helping improve this manuscript through their feedback.
>
> Answers to questions:
>
> Identical masks at initialisation/convergence: Within the sparse neural network community it has previously been observed that masks chosen via magnitude-based pruning at initialisation which are kept fixed throughout training lead to significantly worse results (this is the method "Static" in Table 1). It is thus desirable to maintain some amount of change between initial and final masks to avoid the “static" behaviour. As the reviewer correctly explains, it is intuitive to assume that powerpropagation will decrease the amount of change that happens between masks. The key question now is whether good performance is still possible despite this decreased amount of change: The answer for Powerpropagation is yes! Thus, this suggests that there is a "sweet spot" between no change to the masks ("static") and the change to masks we see for standard learning. The empirical results suggests that Powerpropagation moves us much closer to this spot, which Figure 1b) shows. Finally, to clarify: We never observed identical masks for powerpropagation during all experiments and thus avoid the "static" behaviour which would be undesirable.
>
> Permuted MNIST architecture: This is a ReLU network with two layers of 1024 units and a single output layer shared between all tasks.
>
> Split CIFAR 100 experiments: We performed the requested 10-task Split CIFAR-100 experiment following the protocol in Gradient Projected Memory for Continual Learning (Saha et al, 2021). The results are:
>
> - Multitask Learning (Upper bound): 79.58
> - Elastic Weight Consolidation (Kirckpatrick et al., 2016): 68.80
> - Orthogonal Weight Modulation (Zeng et al., 2018): 50.94
> - A-GEM (Chaudhry et al., 2019): 63.98
> - Experience Replay + Reservoir Sampling: 71.73
> - HAT (Serra et al, 2018): 72.06
> - Gradient Projected Memory (Saha et al, 2021): 72.48
>
>
> - Powerprop. (alpha=1.625) + EPN: 74.16
>
> To the best of owner knowledge, these are new state-of-the-art results for this benchmark.
>
> Efficiency of Exponentiation in Resource-constrained devices: We are not entirely sure about this question. The common way to compare computational cost regardless of the device are Floating Point Operations (FLOPs). In Table 2, we show that Powerpropagation leads to superior performance at reduced training cost (square brackets, FLOPs relativ to cost of training a dense model). Furthermore, resource-constrained devices themselves are unlikely to be used for training and tend to be primarily used for inference. In this case, the Powerpropagation. exponentiation can simply be computed once (as weights don’t change anymore), stored and then directly applied. In this case, the algorithm will lead to better results at equivalent inference cost, as we show throughout the experiments. In case these points do not answer the question, could the reviewer kindly explain in which cases this analysis does not hold?

---

### Official Review · Reviewer_o3yJ · 2021-07-28

**Rating:** 7
**Confidence:** 4

**Summary:**

Proposes a reparameterization of neural network models, where each weight is raised to the power of alpha, where alpha > 1, but keeps its original sign.
The proposed reparameterization and its implications on the optimization process are briefly discussed mathematically, with the general intuition being that "the rich get richer", i.e., low-magnitude weights tend to stay close to zero, while high-magnitude ones tend to become more significant in the resulting model.

Thus, the reparameterization implicitly acts as a regularizer that prefers models with few high-magnitude weights. The resulting model is more suitable for post-pruning schemes that create sparse models.

Finally, the reparameterization is empirically shown to enhance a Continual Learning scheme that uses iterative pruning (PackNet [Arun and Lazebnik 2017]), improving results on various continual learning baselines.

**Ethical Concerns:**

I believe there are no such issues.

**Limitations And Societal Impact:**

I believe the authors addressed these issues adequately.

**Main Review:**

* Generally, the general idea is interesting and extends previous ideas in the field of pruning and sparse models.
* The paper is mostly well written and ideas are presented clearly (the plots are great).
* Empirically, there are decent results in various settings (pruning for sparse models; and implications in an iterative pruning model for continual learning).
* I have some reservations, mostly regarding some unclear paragraphs or sentences.
* I am ambivalent about the rating, **unsure between 6 and 7**. I believe the author response can help in this regard.

---

### A. Originality and significance


1. **Originality.** I believe the proposed reparameterization is original *enough*.
Similar reparameterizations were previously studied in the sparse context, e.g. the authors here adequately cite [Vaskevicius et al. 2019] that reparameterize a linear regression model as $u\circ u- v\circ v$, which at a certain initialization scale limit converges to a minimum-L1 solution (inducing sparsity, see also [Woodworth et al. 2020]).
However, the reparameterization in the paper reviewed here is more *general and flexible* (the exponent can be any $\alpha>1$ and moreover preserves the signs of the exponentiated weights, thus there is no need to split the parameters into $u$ and $v$).

2. **Significance.** The results are interesting to me and seem important. The reparameterization is rather elegant, can evidently yields sparser models. I believe it is immediately (and easily) applicable to general sparse schemes and more specifically pruning schemes. The authors show (Table 1) how it improves several pruning schemes and advances SOTA algorithms on sparse ResNet-50 for ImageNet.
The observations on the continual learning setting are interesting but limited, since the paper focuses on a setting where the task identities are known during inference, and it seems that some of the conclusions only apply to models like PackNet that are based on iterative *pruning*, and do not apply to the wide continual learning scope.

---

### B. Quality and clarity
Most of the paper is clearly written and overall the paper has a good flow. I also believe the experimental results are strong enough.
However, some of the theoretical analysis/intuitions are not really clear to me, and there are a few paragraphs that would benefit from rephrasing. Moreover, I was left with some questions and remarks I would like to specify.

1. I could not understand what exactly do the authors claim in lines 121-124. Please clarify.
2. I could not understand what is the difference between plain backpropagation and what the authors refer to in lines 136-145. This paragraph was extremely unclear to me. Could the authors please clarify? (I believe this paragraph needs to be improved).
3. The description of Algorithm 1 in the beginning of Section 4 is not self contained (some parts cannot be understood without reading the PackNet paper). For instance, it is not clear enough that this setting assumes the task identities are known during inference (thus it was unclear to me while reading why do you store *all* forward masks and not only the current one and the backward one). I also believe it should be explained that the space complexity requirement is valid for both train and inference. Moreover, I could not understand the exact origin of the sparsity range in $S$ in the algorithm. How is it determined?
4. Should the line 12 in Algorithm 1 be split into two lines? This is confusing.
5. Figure 2c: which dataset and model is presented here? CIFAR or ImageNet? Not stated.
6. In line 208: should it really say $s_n\cdot P$? If so, why $s_n$?
7. In line 212-213: the authors state that the algorithm "overcomes" the assumption of known task number. Is it actually true? Shouldn't one at least know the order of $T$ to set the sparsity ranges?
8. In Section 6.3 (continual learning experiments). Is the comparison to EWC (for instance) fair?
Since EWC doesn't assume known task-identities during inference like EPN does.
9. Do any of the experiments use explicit regularization (weight decay)? Could not find the answer in the paper or Appendix.
I guess the answer is no, but I want to make sure. This should also be explicitly written in the paper.
10. It can be interesting to understand or discuss the layer-wise effect of Powerpropagation. For instance, perhaps earlier layers are more dense while later layers are more sparse, etc.

---

### C. Minor remarks
Following are some minor remarks by line numbers. Please consider fixing them.
* 52: occupy => occupies
* 100 (and more): I don't believe the $\circ$ in the exponent was defined.
* Figure 1: should the caption say $\alpha=3.0$ instead of $4.0$?
* 188: Percent sign seems unnecessary
* Algorithm 1, line 12: Should the index in the superscript be $i$?
* 221, 225, 231: Spare => Sparse
* 223: Throughout what?
* 263: **D**atasets should be lowercase.
* 284: shows => show
* 289: constitues => constitute
* 300: significant => signficantly
* 303: Do you actually mean the *gradient* flattens? Or that the plot flattens?
* Generally: Instead of writing "details in the Appendix", consider writing the exact section of the appendix.

* Table 4: Isn't PackNet's forgetting better than EPN? A negative forgetting is often called "backward transfer" and is a good property of an algorithm.

**Time Spent Reviewing:**

12

---

> ### Author Response · Authors · 2021-08-09
> **Response to reviewer o3yJ**
>
> We would like to express particular gratitude to the reviewer for their outstanding attention to details and help in finding various typos and smaller errors. We will correct these in the camera-ready version. Answers to questions:
>
> Question 1: A possible concern for Powerprop. is that standard Deep Learning initialisation (chosen to maintain equal activation variance throughout the network based on the forward pass formulation) now becomes sub-optimal, as the forward pass changes. In order to overcome this, we can modify the initialisation such that an untrained Powerprop. network makes identical predictions to an untrained standard network initialised with the same weights, despite the difference in the forward pass. This ensures that early on in training, the Powerpropagation network should be as amenable to gradient-based learning as a standard network. Note that this trick is also implemented in the accompanying code, which may further improve clarity.
>
> Question 2: Most modern Deep Learning optimisers (e.g. Adam, RMSProp) scale the update by multiplying the gradient with 1/sqrt(\hat{g}^2) where \hat{g}^2 is a running average of the past gradients (squared). The concern is that the Powerpropagation update factor (\alpha*|\phi|^{\alpha-1} [eq. (1)]) now appears in both the numerator and denominator, thus cancelling and "undoing" the powerpropagation scaling and reverting to standard learning. The trick in lines 136-145 overcomes this issue. (Note that the cancelling would only be approximate, as \phi changes between each update, nevertheless this effect does happen as we show in the appendix.)
>
> Question 3: We agree with the reviewer that this algorithm is not self contained and will improve its presentation and clarify its assumptions.
>
> Regarding the assumption of known tasks ids, it is actually possible to overcome this problem. To do so we combined our method with a mask inference algorithm that can be applied at test time (“Supermasks in Superposition”, NeurIPS 2020). The algorithm selects the subnetwork (given by a specific mask) by arguing it should have low entropy (i.e. be confident in its prediction), thus an objective that can be computed in an unsupervised fashion (i.e. at test time) and differentiated. We were able to correctly identify the correct mask at test time in all cases for the Permuted MNIST/Split MNIST (see Appendix) and RL experiments, i.e. performance was unchanged when we relaxed the known task ids at inference assumption. We aim to include these in the camera-ready version.
>
> How is sparsity range determined? In its current form the sparsity range is determined by a user and can be chosen to keep the overall cost of the procedure low (i.e. to satisfy a fixed budget of validation set evaluations). As this assumes the user has an intuition of what sparsity values might be most appropriate, we acknowledge that this may be an unrealistic assumption. We will thus include binary search for up to k steps (thus log(k) complexity) as an automated alternative. The user will thus only be required to specify an acceptable loss in performance per task (\gamma) and a maximum number of search steps k.
>
> Question 4: Agreed. We will improve this.
> Question 5: This is ImageNet. We will clarify this. Thank you.
> Question 6: This should say s_n. Once again, thank you for your attention to details.
>
> Question 7: See our response to question 4. With this modification, the user merely needs to specify an acceptable decrease in performance (likely to be given by system requirements) as well as a maximum number of steps k, but know maximum number of tasks or sparsity range.
>
> Question 8: See answer to Question 4 regarding known tasks ids. More generally: Unfortunately CL methods tend to often make fairly different assumptions thus making a 1:1 comparison often impossible in practice. EWC (in its original formulation in (Kirckpatrick et. al, 2016)) in particular assumes that a the full set of network parameters can be stored after each task, which we don't require. The original work also uses task-specific biases (requiring a task-id at test time), although this is often no longer done in practice. Other CL algorithms we compare against assume access to past data or gradient storage. The research community seems to have concluded that such benchmark comparisons are nevertheless useful despite the varying assumptions. However, we fully agree with the reviewer that it is important to make such differences clear to provide transparency and will provide a detailed list of assumptions for each algorithm in the appendix.
>
> Question 9: Yes, the ImageNet experiments all use weight decay (see Table 7 in the Appendix). We will make this clear in the main text. We also think that a more careful study of the interaction of weight decay and powerprop could be fairly interesting and add to the paper. We will attempt to include this.
>
> Question 10: Thank you for this suggestion. Thank you for this suggestion. We investigated this in the scope of the CIFAR-10 experiment and did indeed observe a difference in sparsity across layers (for an identical overall budget) for varying alpha. The overall observation suggested that increasing alpha leads to a lower sparsity for early convolutional layers and higher sparsity for later dense layers (relative to the alpha=1.0 baseline) which may be an effect of the variance in the initialiser distribution (a function of the layer sizes). In short: Lower initialiser variance may lead to sparser layers, as weights are closer to zero.

---

> > ### Comment · Reviewer_o3yJ · 2021-08-15
> > **Increasing my review's rating**
> >
> > Dear authors,
> >
> > After reading the other reviews and your responses, I believe that a rating of 7 better fits your manuscript.
> > Below are some of my comments on your answers to my review.
> >
> > Good luck!
> >
> > * Answer 1: I understand it now but I cannot see how it can be clear from the current paragraph. Please consider clarifying this in the main text or in the appendix.
> > * Answer 2: OK, consider elaborating in the appendix (and referring to the exact section in the appendix).
> > * Answer 3: OK. Regarding the task IDs, I did not mean that this assumption is very limiting, but that it was not clear from the text that you make this assumption. Regarding the sparsity range - I don't know if it needs to be overcomplicated like your answer suggests, but whatever you choose need to be explained adequately in the text (currently such an explanation is missing).
> > * Answer 6: Are you sure the correct index is n?
> > * Answer 8: OK.
> > * Answer 9: OK. At least mention it. Studying the interaction can be nice-to-have, but it might also complicate the current paper.
> > * Answer 10: Thank you. Your answer makes sense.

---

> > > ### Author Response · Authors · 2021-08-16
> > > **Response to Reviewer o3yJ**
> > >
> > > Thank you for your response and the update of your score. We'll try our best to address all these points in the camera ready version.
> > >
> > > Answer 6: We'll make this clearer by directly explaining that the actual sparsity after each task is upper-bounded by the maximum sparsity defined in the sparsity range.

---

### Author Response · Authors · 2021-08-31
**Additional results on Language modelling**

Dear all,

We would like to share additional sparsity results on Language Modelling using the popular TransformerXL model. We feel that this helps strengthen the argument that Powerpropagation is truly model-agnostic and general reparametrisation and also ensures that the experimental evaluation is comparable to recent work in the area (as both the referenced RigL and TopKAST include language modelling results). All results show TransformerXL on the enwik8 dataset. We report the bit-per-character (BPC) metric on the test set.

To be added to section 6.1 (inherent sparsity):
One-shot pruning: https://drive.google.com/file/d/1JjNMgqy4rY_cELyFGW53kVgjVN9Ag5CX/view?usp=sharing

To be added to section 6.1 ( Advanced Pruning):
Iterative pruning: https://drive.google.com/file/d/19MSrdwtLCU3Px-HiqyshzeyY_bqDDH4W/view?usp=sharing

Thank you!

---

### Decision · Program_Chairs · 2021-09-27

**Decision:**

Accept (Poster)

**Comment:**

This paper suggests a simple re-parameterization, motivated by recent theoretical results, which for improves SOTA results in the areas of pruning and continual learning. All reviewers seemed to like the idea, the writing and the results. The rebuttal addressed remaining concerns, and all reviewers voted for acceptance.